# Ex Situ Conservation of *Atriplex taltalensis* I.M. Johnst. via In Vitro Culturing of Its Axillary Shoots

Carolina Muñoz-Alcayaga [1], Jorge Soto [1], Celián Román-Figueroa [1,2] and Manuel Paneque [3,*]

1 Bionostra Chile Research Foundation, Almirante Lynch 1179, San Miguel, Santiago 8920033, Chile
2 Doctoral Program in Sciences of Natural Resources, Universidad de La Frontera, Av. Francisco Salazar 01145, Temuco 4811230, Chile
3 Department of Environmental Sciences and Natural Resources, Faculty of Agricultural Sciences, Universidad de Chile, Santa Rosa 11315, La Pintana, Santiago 8820808, Chile
* Correspondence: mpaneque@uchile.cl; Tel.: +56-22-978-5863

**Abstract:** *Atriplex taltalensis* is an endangered shrub endemic to northern Chile. Sparse populations of this species can be found in areas with extreme edaphoclimatic conditions in the Atacama desert, and its seeds have a poor germination rate. Consequently, mass-cultivating it can be challenging. In this study, the vegetative propagation mechanisms of *A. taltalensis* were evaluated using an in vitro culture to aid in their conservation. *A. taltalensis* shoot explants were treated with two phytoregulators, indole-3-butyric acid (IBA) and 6-benzylaminopurine (BAP), to assess the morphogenic responses and their effects on the propagation of the species, based on shoot multiplication, elongation, and rooting, through subculturing. During multiplication, the treatment with IBA alone efficiently promoted explant elongation, lateral root formation, and axillary shoot proliferation, allowing for the rapid development of shoots into whole plants. Alternatively, treatment with IBA and BAP in combination stimulated the proliferation of basal shoots with little elongation and rooting and promoted shoot hyperhydricity at 0.25–1 mg L$^{-1}$ BAP concentrations. Thus, we conclude that *A. taltalensis* propagation is viable through in vitro plant tissue culture using a rapid axillary shoot multiplication system, and this method could aid in the conservation of this species through in vitro propagation and rescue programs.

**Keywords:** arid and semi-arid environments; micropropagation; ex situ conservation; dry land; in vitro culturing; endemic plants



## 1. Introduction

*Atriplex* spp. are perennial woody shrubs, distributed mainly in the arid and semi-arid regions of the world [1,2]. These shrubs grow under extreme environmental conditions that include high temperature, high light intensity, high salinity, and drought [3,4], as well as in low nutrient-content soils [5].

However, the seed propagation of *Atriplex* plants is hindered due to their seed dormancy and short viability period [6,7], with low germination rates ranging from less than 2–7% [4,8]. In addition, their germination is further delayed by high soil salt content [4,9]. Due to the high soil aridity in their habitat, *Atriplex* seeds remain dormant until optimal moisture conditions [9,10]. Rooting of cuttings is an alternative propagation strategy for plants of this genus [9,11,12]; however, that process is time-consuming, requires large amounts of space, and its efficiency varies from species to species [13,14].

*A. taltalensis* is an endemic species of Chile, with a distribution restricted to some areas with coastal influence in the regions of Tarapacá and Antofagasta in northern Chile [15]. *A. taltalensis* is an endangered (EN) species owing to its limited distribution that covers an area less than 5000 km$^2$ (estimated to be less than 2300 km$^2$), with an area of occupancy less than 500 km$^2$ (estimated to be less than 72 km$^2$). Furthermore, the species occurs in only

two localities, and the quality of its habitat has decreased due to herbivory and trampling by animals [16].

*A. taltalensis* grows on gentle slopes and river valleys, strictly under high humidity conditions caused by the daily coastal mist [17,18]. The highest abundance of this species occurs in rainy years, triggered by the increased rainfall resulting from El Niño-southern oscillation events [16]. During the dry years, low rainfall prevents the mass emergence of seedlings, as individual plants are unlikely to reproduce from seeds. Such unfavorable conditions are predicted to increasingly affect the species, according to projections of drought events for northern Chile [19,20].

Currently, no information is available on the mass reproduction of *A. taltalensis* as an ex situ conservation strategy. Nevertheless, in vitro culture by direct organogenesis has been described for *A. nummularia* Lindl. [21], *A. glauca* L. [22], *A. canescens* (Pursh) Nut. [2,23], *A. halimus* L. [3,7,8,24], and *A. semibaccata* R. Br. [8]. Therefore, with the implementation of biotechnological techniques, such as in vitro plant tissue culture [25,26], the large-scale propagation of *A. taltalensis* may be facilitated and can aid in its reintegration and ex situ conservation [27–29].

In vitro plant tissue culture involves isolating organs, tissues, or cell explants from a donor plant and adjusting the necessary conditions to induce physiological or morphogenic responses [25,30]. In *Atriplex*, different morphogenic responses have been evaluated, including bud and shoot multiplication, shoot elongation, rooting, and whole plant regeneration, using different initial explants such as nodal segments [8,21,24], flower tissues [22], shoot tips [3,8], and axillary buds [3,7]. In addition, the ability of various cytokinins, namely, 6-benzylaminopurine (BAP), 2-isopentenyladenine (2iP), thidiazuron, kinetin (KIN), and zeatin (ZEA), to promote the in vitro multiplication of *Atriplex*, has been tested [3,7,23,24].

In *Atriplex*, in vitro elongation and rooting have also been studied using the auxins indole-3-acetic acid (IAA), indole-3-butyric acid (IBA), naphthaleneacetic acid (NAA), and gibberellic acid (GA3) [3,7,8,23]. IBA is the main in vitro rooting promoter of woody plants [31]. Furthermore, both cytokinins and auxins have promoted in vitro *Atriplex* differentiation and growth when used alone or in combination [3,7,8,21].

However, in vitro *Atriplex* propagation for whole-plant regeneration is limited due to excessive bud growth, with little shoot elongation, and without differentiation [23]. In addition, bud and shoot hyperhydricity range from 30–90% [2], which induces a physiological disorder that is common in in vitro cultures and disrupts the anatomy and physiology of plants, in turn affecting their establishment and rooting [32].

Considering that *A. taltalensis* is an EN species, and no information on its reproduction and conservation is available, a method for its mass and short-term propagation is imperative. Accordingly, in the present study, in vitro *A. taltalensis* propagation was evaluated, and the morphogenic response in shoot explants, namely their multiplication, elongation, and rooting, to different treatments with phytoregulators was determined. The results of the present study could facilitate the propagation and conservation of the species.

## 2. Materials and Methods

### 2.1. Disinfection of Plant Material and Collection of Initial Explants of A. taltalensis

Portions of lateral branches of *A. taltalensis* plants were collected, complying with applicable regulations, from the Taltal commune, Antofagasta Region, Chile. The plant materials were stored in polyethylene bags and then transferred to the laboratory in a refrigerated container at 2–8 °C to keep them fresh until use [33]. The branches were superficially disinfected by immersing them in 80% (*w/v*) Captan solution for 20 min. Subsequently, under aseptic conditions in a laminar flow cabinet, the branches were further immersed in 70% (*v/v*) ethanol for 30 s [34], followed by 0.1% (*w/v*) mercury (II) chloride for 5 min [35]. At the end of each disinfection step, the branches were rinsed three times with distilled water [35]. After the branches were disinfected, 1.5 cm long single-node segments of the branches with a visible bud in the node portion were selected and cut (Figure 1A). The segments were established in Murashige and Skoog (MS) solid medium [36] for 28 days until

shoot emergence. The shoots were then used as initial explants for in vitro multiplication of *A. taltalensis*.

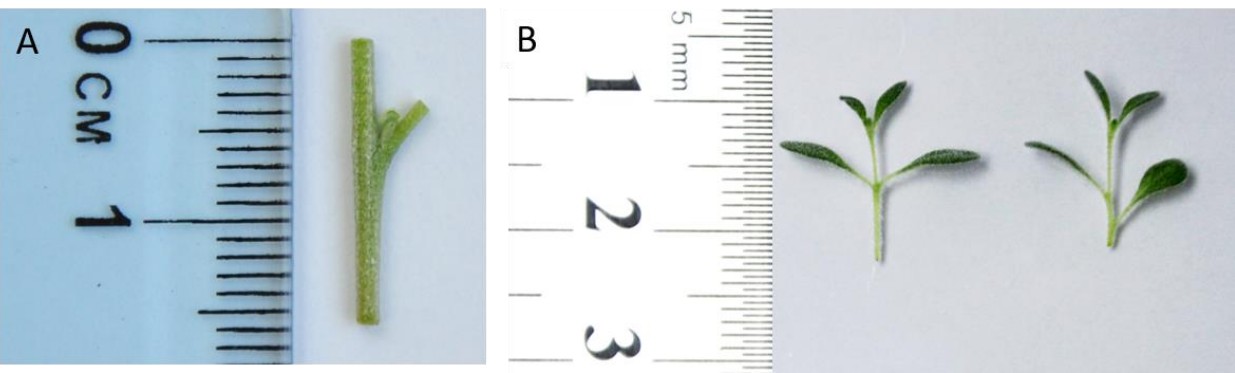

**Figure 1.** Propagation of initial explants from plant collections: (**A**) a uninodal segment used for the in vitro establishment of *A. taltalensis*; (**B**) Initial explants with shoots used for multiplication.

### 2.2. Culture Conditions

#### 2.2.1. Culture of Initial Explants of *A. taltalensis* under Different Multiplication Treatments

The 1–2 cm long shoot initial explants with 1–2 nodes (Figure 1B) were excised from the plants grown in vitro and vertically cultured for 45 days in 100% MS basal medium and 3% ($w/v$) sucrose, supplemented with 0.5 mg L$^{-1}$ IBA alone or in combination with different concentrations of BAP (0.05, 0.25, 0.50, and 1 mg L$^{-1}$) (Table 1). The explants were cultivated in glass flasks (7.5 × 20 cm) with 30 mL of culture medium and then covered with a polypropylene lid. The flasks were then kept in a growth room based on the required biosafety standards at 22 ± 2 °C, a 12-h photoperiod with 70.16 µmol m$^{-2}$ s$^{-1}$ light intensity, and 12 h of darkness. At the end of day 45, a qualitative observation was performed to compare the morphogenic responses of the explants to the treatments. The treatments were evaluated in terms of the number of shoots formed per explant ($n$), the percentage of explants that produced shoots, and the percentage of explants that formed roots. All explants with an apex and with 1–3 nodes were considered shoots.

**Table 1.** Treatments tested for the in vitro multiplication of *A. taltalensis* during the initial explants culture and shoot subculture.

| Culture Stage | Treatment | Basal Medium + Phytoregulators |
|---|---|---|
| Multiplication | AEmc * | MS |
| | AEm5 | MS + 0.50 mg L$^{-1}$ IBA |
| | AEm1 | MS + 0.50 mg L$^{-1}$ IBA + 0.05 mg L$^{-1}$ BAP |
| | AEm2 | MS + 0.50 mg L$^{-1}$ IBA + 0.25 mg L$^{-1}$ BAP |
| | AEm3 | MS + 0.50 mg L$^{-1}$ IBA + 0.50 mg L$^{-1}$ BAP |
| | AEm4 | MS + 0.50 mg L$^{-1}$ IBA + 1.00 mg L$^{-1}$ BAP |

* Control medium. MS: Murashige and Skoog medium (1962); IBA: indole-3-butyric acid; BAP: 6-benzylaminopurine.

#### 2.2.2. Effect of Shoots Subcultures on the Multiplication, Elongation, and Rooting In Vitro of *A. taltalensis*

The shoots generated in the previous phase were then subcultured twice, every 30 days, in the same treatments, on fresh media (Table 1), to determine the efficacy of the treatments on shoot production [3]. The explants were cultivated under the same conditions used in the initial culture. Each shoot explant was evaluated in terms of the number of new shoots ($n$) formed, shoot length (cm) measured from the base to the apex of the youngest leaf [3,24,37], shoot rooting (%), and hyperhydricity (%). The percentage of hyperhydricity

was assessed by classifying newly formed shoots as normal shoots or hyperhydridic shoots, according to their external appearance [38].

### 2.3. Statistical Analysis

The assumptions of data normality and homogeneity of variance were tested using the Shapiro–Wilk and Levene's tests, respectively [39]. A non-parametric Kruskal–Wallis analysis was performed at a 99% confidence level ($\alpha$ = 0.01) to assess the effects of the treatment on parameters: the number of shoots per explant, shoot length, and percentage of explants that produce shoots and form roots, in the culture of initial explants, and the number of shoots per explant, the height of shoots and percentage rooting, and hyperhydricity of the 1st and 2nd subcultures [40]. Subsequently, post-hoc analysis was performed using Dunn's multiple comparison test, with a critical level of significance of $\alpha$ = 0.01 (1%), to identify statistically significant differences between the results of the treatments evaluated in each culture stage [41,42]. All analyses were made with four replicates, and each replicate included 15 explants in the initial culture and 30 explants in the subculture. These procedures were performed using InfoStat 2020e (National University of Córdoba, Córdoba, Argentina).

## 3. Results

### 3.1. Induction of Initial Explants from Uninodal Segments of A. taltalensis

The growth of the initial shoot explants was induced from the uninodal segments in the cultures. No contamination by microorganisms was observed during the culture of the segments. The initial explants developed from the active bud at the node of the segments achieved adequate growth and size, as well as a vigorous appearance, and were considered the best candidates for the multiplication stage.

### 3.2. Culture Initial Explants

All multiplication treatments promoted basal and axillary shoot proliferation. The treatments AEm2, AEm3, AEm4, and AEm5, which included IBA and BAP (Table 1), stimulated the formation and differentiation of basal buds into shoots, which grew in a cluster at the base of the initial explants (Table 2; Figure 2A–D). The treatments, excluding the AEm2 treatment, failed to promote rooting, caused little elongation, and resulted in non-organogenic callus tissue formation at the base of the initial explants (Figure 2B–D). The basal shoots formed as a result of the treatments typically had leaves with rounded apexes and thick stems and leaves (Figure 2E).

**Table 2.** Morphogenic responses observed in the culture of initial explants of *A. taltalensis* subjected to multiplication treatments.

| Culture Period (Weeks) | Observed Responses |
|:---:|---|
| 1 | Reddish and bulging stem base in explants subjected to treatments AEm2, AEm3, AEm4, and AEm5. |
| 2 | Callus growth in explants subjected to treatments AEm2, AEm3, AEm4, and AEm5. Initial development of lateral roots in explants subjected to the control treatment AEmc and treatments AEm2 and AEm1. Apex elongation and stem growth in explants subjected to treatment AEm1. |

**Table 2.** *Cont.*

| Culture Period (Weeks) | Observed Responses |
| --- | --- |
| 3 | Development of basal buds in explants subjected to treatments AEm2, AEm3, AEm4, and AEm5. Development of axillary buds in nodes of explants subjected to treatment AEm1. Apex elongation and stem growth in explants subjected to treatment AEm2 and control treatment AEmc. |
| 4 | Lateral root growth and proliferation in explants subjected to treatments AEm2 and AEm1. Differentiation of basal buds into shoots in explants subjected to treatments AEm2, AEm3, AEm4, and AEm5. Differentiation of axillary buds into shoots in nodes of elongated explants subjected to treatment AEm1. Development of axillary buds in explants subjected to control treatment AEmc and treatment AEm2. |
| 5 | Callus growth and basal shoot proliferation in explants subjected to treatments AEm2, AEm3, AEm4, and AEm5. Axillary shoot growth and proliferation in nodes of explants growing under control treatment AEmc and treatments AEm2 and AEm1. |
| 6 | Basal (treatments AEm2, AEm3, AEm4, and AEm5) and axillary shoot growth (control treatment AEmc and treatments AEm2 and AEm1). Formation of whole plants subjected to control treatment AEmc, treatments AEm2 and AEm1. |

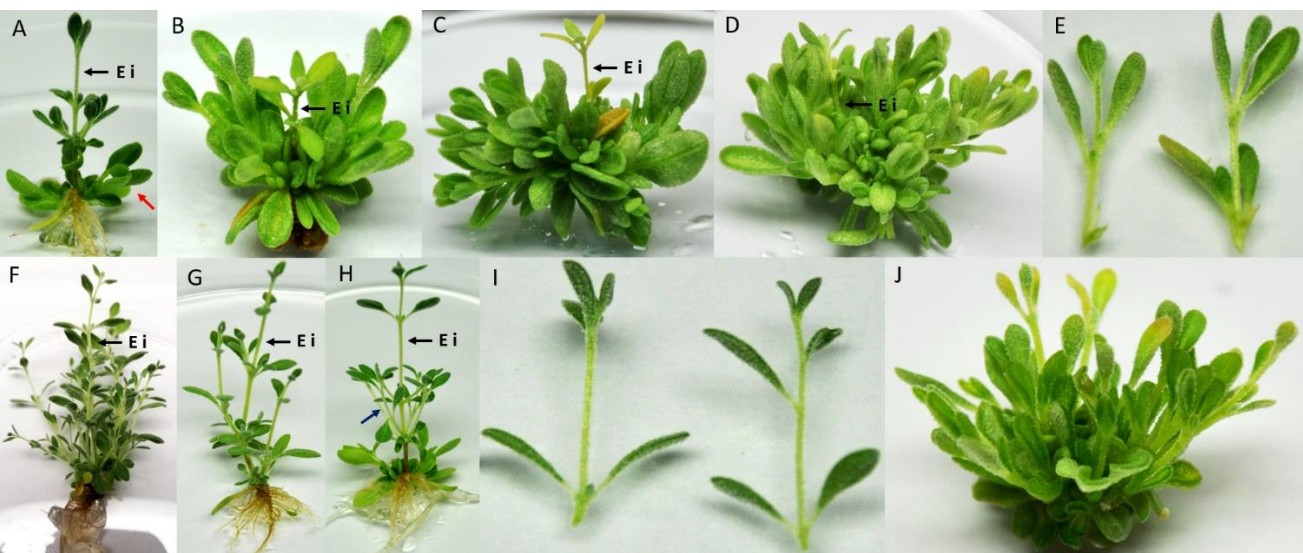

**Figure 2.** *A. taltalensis* shoots grown in vitro under different multiplication treatments: (**A**); (red arrow) Basal shoots growing in clusters from the base of initial explants in AEm2; (**B**) AEm3; (**C**) AEm4; (**D**) AEm5; (**E**) Morphological aspect of the basal shoots; (**F**) Axillary shoots and whole rooted plants growing from initial explants in AEm1; (**G**) AEmc; (**H**); (blue arrow) AEm2; (**I**) Morphological aspect of the axillary shoots; (**J**) Hyperhydrated basal shoots subjected to treatment AEm5 during the 2nd subculture; (Ei) Initial explant.

The control treatment AEmc (without phytoregulators) and treatment AEm1, which included IBA alone (Table 1), stimulated explant apex elongation and axillary bud differentiation into shoots at the stem nodes of developing plants (Table 2; Figure 2F,G). Treatment AEm2, which had the lowest concentration of BAP (Table 1), promoted responses similar to those of the control treatment AEmc and treatment AEm1, albeit weaker (Figure 2H). In the control treatment AEmc and treatments AEm2 and AEm1, the shoot explants exhibited

rapid elongation and lateral root formation between 14 and 21 days of culture, resulting in the development of whole plants at the end of the culture period while the plants subjected to treatment AEm1 were the most physically vigorous (Table 2; Figure 2F–H). The axillary shoots typically had leaves with more defined apexes and thin stems and leaves (Figure 2I). A spontaneous rooting response was observed in the initial explants in the control treatment AEmc, with reduced root formation, when compared to the treatment AEm1.

The treatments AEm4, AEm5, and AEm1 had the highest numbers of shoots per explant, significantly higher than the number of shoots in treatments AEm2 and AEm3 and the control treatment AEmc ($p < 0.01$; Table 3). Treatments AEm4 and AEm5 resulted in a high number of shoots per explant, with a higher number observed for treatment AEm5, which included BAP at a high concentration (Table 3; Figure 2C,D). However, the treatments AEm4 and AEm5 led to a low percentage of initial explants that could produce shoots (37–42%; Table 3). The number of shoots per explant in treatment AEm1 was not significantly different from that in treatments AEm4 and AEm5 ($p > 0.01$; Table 3). However, the percentage of initial explants that produced shoots (65%) was significantly higher in treatment AEm1 than that in treatments AEm4, AEm5, and AEm3 ($p < 0.01$; Table 3). Furthermore, the percentage of explants that formed roots was significantly higher in treatment AEm1 (90%) than that in all the other treatments, including the control treatment AEmc ($p < 0.01$; Table 3).

**Table 3.** Effects of the different multiplication treatments on the growth parameters measured in the initial explants of *A. taltalensis*.

| Treatment | Phytoregulators (mg L$^{-1}$) | N° Shoots/Explant | Explants with Shoots (%) | Explants with Roots (%) |
|---|---|---|---|---|
| AEmc * | Without phytoregulators | 2.57 ($\pm$2.44) [b] | 62.00 [a,b] | 73.00 [b] |
| AEm1 | 0.50 IBA | 4.52 ($\pm$3.94) [a] | 65.00 [a] | 90.00 [a] |
| AEm2 | 0.50 IBA + 0.05 BAP | 2.80 ($\pm$3.51) [b] | 62.00 [a,b] | 37.00 [c] |
| AEm3 | 0.50 IBA + 0.25 BAP | 2.50 ($\pm$3.99) [b] | 38.00 [c] | 0.00 [d] |
| AEm4 | 0.50 IBA + 0.50 BAP | 4.83 ($\pm$6.65) [a] | 42.00 [b,c] | 0.00 [d] |
| AEm5 | 0.50 IBA + 1.00 BAP | 5.34 ($\pm$7.90) [a] | 37.00 [c] | 0.00 [d] |

*: control medium. IBA: indole-3-butyric acid; BAP: 6-benzylaminopurine. Different letters in a column indicate significant differences according to Dunn's test ($p < 0.01$). Data correspond to the mean of four replicates of 15 repetitions. SD: standard deviation.

### 3.3. Subculture of New Shoots for In Vitro Multiplication

The subculture of shoots under all the multiplication treatments resulted in proliferation, elongation, and rooting responses, similar to those observed in the initial explant cultures (Table 2). However, subcultures under different multiplication treatments showed differences in the rooting response and the hyperhydricity effect, particularly in treatments with the highest concentrations of BAP ($\geq$0.25 mg L$^{-1}$).

In the 1st and 2nd subculture, treatment AEm1 resulted in a higher number of shoots per explant, compared to the remaining treatments and the control AEmc ($p < 0.01$; Table 4). Axillary shoots subcultured in treatment AEm1 displayed rapid stem growth and node development with growing buds and stimulated the formation of five to six new axillary shoots per explant, on average. The longest shoots, averaging 1.35 cm, were identified for explants subjected to treatment AEm1, albeit with no significant difference from the mean shoot length in treatments AEm4, AEm5, and the control treatment AEmc ($p > 0.01$; Table 4). In contrast, the AEm3 treatment had the lowest number of shoots per explant and the lowest length observed (Table 4).

**Table 4.** Effect of the different multiplication treatments on the growth parameters measured in *A. taltalensis* shoots after subculture.

| Treatment | Phytoregulators (mg L$^{-1}$) | N° Shoots/Explant | Shoot Height (cm) | Rooting (%) | Hyperhydricity (%) |
|---|---|---|---|---|---|
| | | 1st subculture | | | |
| AEmc | Without phytoregulators | 2.06 (±1.90) [d] | 1.25 (±0.38) [a,b] | 74.00 [a] | 0.00 [a] |
| AEm1 | 0.50 IBA | 6.25 (±3.68) [a] | 1.36 (±0.48) [a] | 89.00 [a] | 0.00 [a] |
| AEm2 | 0.05 BAP + 0.50 IBA | 2.85 (±2.44) [c,d] | 0.94 (±0.25) [c,d] | 34.00 [b] | 0.00 [a] |
| AEm3 | 0.25 BAP + 0.50 IBA | 2.00 (±1.38) [d] | 0.86 (±0.21) [d] | 15.00 [b,c] | 31.00 [b] |
| AEm4 | 0.50 BAP + 0.50 IBA | 3.82 (±2.50) [b,c] | 1.17 (±0.27) [a,b,c] | 6.00 [c] | 27.00 [b] |
| AEm5 | 1.00 BAP + 0.50 IBA | 4.92 (±3.68) [b] | 1.11 (±0.27) [b,c] | 5.00 [c] | 24.00 [b] |
| | | 2nd subculture | | | |
| AEmc | Without phytoregulators | 2.08 (±2.11) [c] | 1.21 (±0.40) [a,b] | 59.00 [b] | 0.00 [a] |
| AEm1 | 0.50 IBA | 5.94 (±3.50) [a] | 1.35 (±0.40) [a] | 87.00 [a] | 0.00 [a] |
| AEm2 | 0.05 BAP + 0.50 IBA | 2.72 (±3.02) [b,c] | 1.01 (±0.28) [b,c] | 43.00 [c] | 0.00 [a] |
| AEm3 | 0.25 BAP + 0.50 IBA | 1.81 (±1.33) [c] | 0.86 (±0.23) [c] | 17.00 [d] | 41.00 [b] |
| AEm4 | 0.50 BAP + 0.50 IBA | 3.18 (±2.61) [b,c] | 1.14 (±0.27) [b,c] | 14.00 [d] | 53.00 [b] |
| AEm5 | 1.00 BAP + 0.50 IBA | 3.60 (±3.52) [b] | 1.16 (±0.28) [b,c] | 13.00 [d] | 43.10 [b] |

Different letters in a column indicate significant differences according to Dunn's test ($p < 0.01$). Data correspond to the mean of four replicates with 30 repetitions. SD: standard deviation.

The highest rooting percentage (89%) was observed for shoots under the AEm1 treatment in the 1st subculture, with no statistically significant difference from that of the AEmc control treatment ($p > 0.01$; Table 4). All roots grown under the treatments AEm1 and AEmc were physically healthy and developed lateral roots. Conversely, shoots subjected to treatments AEm3, AEm4, and AEm5 only showed the development of a few adventitious roots, with the AEm5 treatment having the lowest percentage of rooted shoots (Table 4).

Basal shoots of the 1st and 2nd subculture under treatments AEm3, AEm4, and AEm5 showed tissue hyperhydricity, and were characterized by the presence of stems and leaves with increased thickness and vitreous appearance, compared to the treatments AEm1 and AEm2, and the control AEmc (Figure 2J). Shoots with hyperhydricity induced the proliferation of new shoots with the same condition without organ malformations. The shoots of the 2nd subculture grown under treatments AEm4 and AEm5, with the highest BAP concentration, had significantly higher numbers of hyperhydric shoots (>40%) than the rest of the treatments and the control AEmc treatment ($p < 0.01$; Table 4), which had the lowest BAP concentration or no BAP application.

## 4. Discussion

Morphogenic responses, including basal and axillary shoot growth, in the initial explants under different multiplication treatments, were crucial for the efficient propagation of *A. taltalensis* during the subculture process. These responses were influenced by the type and concentration of the phytoregulators used, which stimulated different growth patterns.

Treatment AEm1 (0.50 mg L$^{-1}$ IBA), during the initial explant culture and the following subculture, stimulated a rapid elongation response in axillary shoots and the formation of lateral roots and facilitated the proliferation of shoot explants. In addition, the AEm1 treatment triggered the shoot tissue to initiate elongation until the formation of whole plants during multiplication. Such responses are in accordance with previous studies of in vitro plant propagation and have been observed to be strongly affected by the addition of auxins to the culture medium, whose action is essential for plant growth, because they induce elongation, rooting, and differentiation at different times, and in different tissues,

wherein they are differentially distributed [8,43–46], and for the endogenous balance of auxins and cytokinins of the explants in culture [47].

In vitro shoot elongation in explants induced in response to IBA only has not yet been studied in *Atriplex* spp. However, elongation in response to the combinations of the IBA and IAA auxins has been described previously by Terán et al. [8], based on the apical growth of axillary buds in explants of node segments from *A. halimus* and *A. semibaccata*, and stimulated bud elongation, stem growth, and the formation of viable plants. In addition, the authors induced root development in vitro in *A. semibaccata* apexes using only IBA (2 mg L$^{-1}$), whose effect was greater in relation to IAA. In the present study, the efficient rooting response (87–89%) to the AEm1 treatment observed in *A. taltalensis* axillary shoots (Tables 3 and 4) corroborates the findings of Terán et al. [8] and is similar to that reported by Reyes-Vera et al. [2] who demonstrated that 0.5 mg L$^{-1}$ IBA, the IBA concentration in the AEm5 treatment (Table 1), is the most suitable, promoting 88% rooting in internodal cuttings of *A. canescens* shoots in vitro. However, Aldahhak et al. [3], when assessing the effect of 0.5 mg L$^{-1}$ IBA on *A. halimus* shoot tips in vitro, observed a low rooting percentage (20%). According to Terán et al. [8], the effects of auxin treatments may differ across species and explant physiology, age, and genetics are underlying factors.

The rapid shoot elongation response observed under the AEm1 treatment, up until when the plants developed fully, could be regulated by the early formation of lateral roots in shoot explants (14 days; Table 2), owing to the effect of exogenous IBA, which contributes to the rapid uptake of essential macronutrients available in the culture medium, mediated by the roots. Lloret and Casero [48] observed that exogenous auxin application to *Allium cepa* and *Zea mays* root explants stimulates the formation of lateral roots. IBA, and auxins in general, have demonstrated positive effects on plant development by accelerating the formation of lateral roots and enhancing the development of the root system [49–52]. Consequently, the uptake of essential nutrients, such as nitrogen (N), phosphorus (P), and potassium (K), accelerates plant development and improves shoot growth [52,53]. Our results demonstrate that *A. taltalensis* shoots under the AEm1 treatment responded efficiently to low concentrations of exogenous IBA (0.5 mg L$^{-1}$) and demonstrated improved root growth, which favored whole-plant development in a short culture period (35 days; Table 2). Treatment AEm1, which included only IBA, had a positive effect, inducing shoot elongation and the formation of lateral roots in initial explant cultures and shoot subcultures. These responses show that IBA receptors in the cells of the apical, root, and stem meristem of *A. taltalensis* shoots were active, receiving and transducing physiological signals at a concentration of 0.5 mg L$^{-1}$, thereby enhancing the morphogenic effects of this treatment (Table 2; Figure 2F). Exogenous auxins bind to target cell receptors on explants, inducing the expression of genes that mediate the physiological effects of these phytoregulators [46,54]. In treatments AEm3, AEm4, and AEm5, which included IBA at the same concentration and BAP between 0.25 and 1 mg L$^{-1}$, IBA cell receptors were inhibited by BAP, suppressing the effects of the auxin IBA. In addition, in the AEm2 treatment, which contained IBA and had the lowest concentration of BAP (0.05 mg L$^{-1}$), shoot elongation and rooting response were also obtained, since such receptors would be active, facilitating the physiological effects of IBA.

The low elongation response and negative rooting of *A. taltalensis* shoots in the BAP treatments at concentrations ≥0.25 mg L$^{-1}$ (Tables 3 and 4) is directly related to the presence of the phytoregulator in the medium, and its effect as a suppressor of cell elongation and an inhibitor of root growth [55–57]. In such treatments, the effect of BAP was antagonistic to the positive effect of IBA, whose endogenous cytokinin/auxin balance in the shoot tissues was favorable for cytokinin, inducing a proliferative response in the explants [47], resulting in the multiplication of basal shoots in relation to elongation and rooting responses (Figure 2B–D).

It has been previously demonstrated that cytokinin inhibits the formation of lateral roots since they alter or reduce the polar transport of auxins by disrupting the expression of the *PIN* gene that codes for an auxin transporter in the founder cells of the root [58].

The exogenous application of cytokinins, such as BAP, KIN, and ZEA, reportedly inhibits lateral root initiation in *Oryza sativa* by repressing lateral root primordium formation [49], and in *Arabidopsis* by blocking the pericycle founder cells from the root [50].

The combination of exogenous cytokinins and auxins or the use of cytokinins alone, such as BAP, have considerable effects on the regeneration of woody plants [3] and have induced in vitro shoot proliferation responses in different species in the genus *Atriplex* [3,7,8,21]. Nevertheless, in the present study, although shoot proliferation was induced in all multiplication treatments, in the AEm1 treatment, which included only IBA, the auxin promoted the formation of new shoots in each explant in the subcultures more efficiently than the effect of IBA and BAP combined in the AEm3, AEm4, and AEm5 treatments (Table 4). Auxin activity is not isolated, and cytokinins affect auxin morphogenic responses at different times and in different tissues during plant development in vitro [46,47].

Although auxins stimulate stem elongation and lateral root formation, they inhibit the formation and growth of axillary buds by actively participating in apical dominance [59,60]; effects that were also observed in the present study during the first weeks of explant shoot growth in the AEm1 treatment during the initial explant culture (Table 2) and shoot subculture. In contrast, cytokinins are strongly involved in axillary bud initiation and growth and are mostly biosynthesized in meristematic cells of the shoot and root organs [60,61]. Considering the above, along with the increased formation of axillary shoots per explant under the AEm1 treatment, which included only IBA (Table 4), the proliferation response could be conditioned by the endogenous cytokinin content of shoot tissues. These cytokinins are produced at different times during development, initially due to biosynthesis in the meristems of stem nodes, and later due to the levels reached in root meristems in response to the early formation of lateral roots in the shoot. Accordingly, the endogenous cytokinin contents of shoots under the AEm1 treatment were higher than exogenous IBA contents, increasing the cytokinin/auxin ratio, and thereby stimulating axillary bud initiation and shoot growth in cultured explant stem nodes.

Shimizu-Sato et al. [60] indicated that the number of endogenous cytokinins locally biosynthesized in stem nodes is sufficient for the promotion of axillary bud growth and conversion to axillary shoots. Conversely, the cytokinin contents in roots are transported to nodes, promoting axillary bud growth and shoot development [62]. Low auxin levels or auxin depletion in stems have been shown to induce adenosine phosphate-isopentenyltransferase gene expression, which promotes local cytokinin biosynthesis in stem nodes and roots, thereby controlling axillary bud growth and various aspects of plant development [60,63].

The responses of *A. taltalensis* shoots to AEmc treatment were similar to those of the AEm1 (IBA) treatment based on the elongation, rooting, and axillary shoot formation, albeit with a limited number of new shoots per explant (Tables 3 and 4). This mitigated response could be related to the balance of endogenous auxins and cytokinins in initial shoot explants, which may have contained higher levels of auxins than cytokinins in their tissues during the initial phase of development, favoring shoot apex elongation (Table 2) and polar auxin transport towards the base of the stem, thereby stimulating spontaneous rooting through the formation of lateral roots. Axillary shoot formation in the AEmc treatment, as in the AEm1 treatment, may be attributed to a favorable balance of cytokinins owing to their biosynthesis in roots and stem nodes, in turn stimulating axillary bud initiation and shoot growth [60,62].

The spontaneous in vitro rooting of *A. taltalensis* shoots grown under control treatment AEmc has been similarly reported in *A. halimus* [3,7,8]. Papafotiou et al. [8] observed 85% rooting in nodal explants grown in the control medium. Conversely, Aldahhak et al. [3] observed 73.33% rooting in shoot tips. Furthermore, Kumar et al. [7] reported a high rooting efficiency (>73%) in regenerated shoots. Some auxin synthesis sites, such as apices and meristems, when used as initial material in in vitro cultures, show sufficient endogenous concentrations of these phytoregulators and can develop tissues and organs without exogenous phytoregulators [64,65].

The hyperhydricity of basal shoots in the AEm3, AEm4, and AEm5 treatments may be attributed to the high BAP concentrations in the culture medium because the hyperhydricity was more prevalent in treatments with 0.5 and 1 mg L$^{-1}$ BAP (AEm4 and AEm5) in the second subcultures (Table 4). Such hyperhydricity, ranging from 30 to 90%, has also been described for other *Atriplex* spp. using different cytokinins [2]. Through in vitro *A. canescens* multiplication, Tripathy and Goodin [66] obtained vitreous shoots using 0.40 mg L$^{-1}$ BAP, as was observed in the present study in the AEm4 treatment (0.5 mg L$^{-1}$ BAP). Reyes-Vera et al. [2] induced hyperhydricity in the shoots of the *A. taltalensis* using a high 2iP concentration (5 mg L$^{-1}$) and obtained buds with suppressed apical dominance and abnormally thick, translucent, and vitreous leaves, matching the morphological responses observed in this study under the same conditions. Cytokinins, such as BAP, induce hyperhydricity in many plant species cultured in vitro, generally in a phytoregulator concentration-dependent manner [38,67]. Successive explant subcultures in media containing cytokinins induce a similar disorder as well [38,68].

The endemism of *A. taltalensis* in northern Chile makes it a unique species, being geographically restricted to only two regions, and specific sectors associated with low slopes and ravine bottoms influenced by coastal fog [17,18]. It is in danger of extinction, mainly due to its limited distribution, sparse population, and the quality and degradation of its habitat due to herbivory and animal transit [16]. Being a species adapted to the extreme edaphoclimatic conditions of the Atacama Desert, it has great survival potential under scarce water resources [69,70]. However, the species requires rainy years for the emergence of copious seedlings, due to the low reproduction potential of seeds [16]. Considering the latter, it is at great risk due to the unfavorable climate change projections for its habitat [19,20].

Despite the importance of *A. taltalensis*, its conservation challenges and the existing climate change threat [16], there are no methods for the short-term generation of new individuals on a large scale to facilitate its protection and repopulation. Consequently, it is essential to study the mechanisms that could facilitate ex situ conservation and the preservation of vulnerable species at risk of extinction to preserve their genetic material [71]. Therefore, in vitro propagation helps to conserve a large number of individuals of the species in small spaces, having greater control over the phytosanitary status of the plants [72]. It also helps to generate biological material, either for in situ repopulation programs, collections in botanical gardens, or material for carrying out applied scientific studies, without the need to collect natural material, considering the need to conserve the scarce local populations [73]. However, it could present some limitations, such as somaclonal variation, with respect to the mother plant [72].

The in vitro methods developed in the present study, through the propagation of the axillary shoots of *A. taltalensis* and exclusively using the phytoregulator IBA at a concentration of 0.5 mg L$^{-1}$, could facilitate obtaining complete and viable plants at a large scale, with an adequate root system, rapidly, which makes them optimal plants for the next stage of the present study, acclimatization. The method could be applied as a strategy for efficiently obtaining new *A. taltalensis* individuals, and in the ex situ conservation and reintegration of the species.

## 5. Conclusions

In the present study, 0.5 mg L$^{-1}$ IBA stimulated explant elongation and lateral root development, enhancing axillary shoot proliferation and the simultaneous formation of whole and viable plants, thereby enabling the rapid in vitro propagation of *A. taltalensis*. Moreover, IBA induced a higher number of axillary shoots per explant (5–6 shoots), longer shoots (1.35–1.36 cm), and a higher rooting percentage (87–89%) than those observed in other treatments.

BAP, at concentrations above 0.05 mg L$^{-1}$, suppressed the positive effect of IBA in multiplication treatments and primarily induced the proliferation of basal shoots, with low

elongation and limited root development. At higher concentrations, BAP caused shoot hyperhydricity, a condition that persisted in subcultures.

The in vitro culture method developed in the present study facilitated the mass and short-term multiplication of *A. taltalensis* plants via axillary shoot propagation using only exogenous IBA. As such, this method may be applied as a conservation strategy for *A. taltalensis*, an EN endemic species. It also offers an opportunity to massively produce plants in a controlled manner, for repopulation purposes, if required, in the context of climate change.

**Author Contributions:** C.M.-A.: conceptualization, methodology, investigation, writing—original draft preparation, visualization. J.S.: conceptualization, formal analysis, writing—original draft preparation, writing—review and editing. C.R.-F.: conceptualization, methodology, validation, writing—review and editing. M.P.: conceptualization, methodology, validation, resources, writing—review and editing, funding acquisition. All authors have read and agreed to the published version of the manuscript.

**Funding:** This work was funded by Agroenergía Ingeniería Genética S.A.

**Acknowledgments:** The authors acknowledge Marely Cuba Diaz for her manuscript revision.

**Conflicts of Interest:** The authors declare no conflict of interest.

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
