# Peer review of "Ex Situ Conservation of Atriplex taltalensis I.M. Johnst. via In Vitro Culturing of Its Axillary Shoots"

_diversity, doi:10.3390/d15010013_

Round 1
Reviewer 1 Report
The content of this research is relatively simple, and need to be further streamlined and standardized. For example, it is important to write concisely about the tissue culture conditions; the BAP treatment should be used in this experiment. Therefore, I don’t think it is appropriate at its current form.
Author Response
We thank the reviewer for their comments. The manuscript has benefited from these insightful suggestions. Response to the reviewer comments is included in an attached file

Reviewer 2 Report
The manuscript entitled "Ex situ conservation of Atriplex taltalensis via in vitro culturing of its axillary shoots" reported the research using uninodal segments of Atriplex taltalensis to produce initial explants from shoot, and then conducted subculture for multiplication. The experiment included multiple treatments with multiple replications included. Data analysis was correctly done, and conclusions were clear. I would recommend publishing the manuscript after some minor revisions.
The section of results would be the part I recommend authors to revise:
1) the secondary title 3.1 and the tertiary title 3.1.1 and 3.1.2 all cause confusions. I would suggest removing the secondary title 3.1 but put three secondary titles in the results as:
3.1. Induction of initial explants from uninodal segments of Atriplex taltalensis. (Did all segments produce good initial explants?)
3.2. Culture initial explants
3.3. Subculture new shoots for in vitro multiplication
Other suggestions:
lines 152 -156 need to remove since the paragraph doesn't talk about results but just basic description.
Lines 189 - 192 need to reword and it has some misleading.
Author Response

(The authors gave the same response as above.)

Reviewer 3 Report
The article is well written and clearly laid out in the appropriate sections. The study in interesting with potential for using the appropriate concentration of IBA for other species that are endangered (EN).
Author Response

(The authors gave the same response as above.)

Round 2
Reviewer 1 Report
I have made my decision, and don't need to made a second time.